# Antibiotic Use: A Cross-Sectional Study Evaluating the Understanding, Usage and Perspectives of Medical Students and Pathfinders of a Public Defence University in Malaysia

**DOI:** 10.3390/antibiotics8030154

**Published:** 2019-09-19

**Authors:** Mainul Haque, Nor Azlina A. Rahman, Judy McKimm, Massimo Sartelli, Golam Mohammad Kibria, Md Zakirul Islam, Siti Nur Najihah Binti Lutfi, Nur Syamirah Aishah Binti Othman, Shahidah Leong Binti Abdullah

**Affiliations:** 1Faculty of Medicine and Defence Health, Universiti Pertahanan Nasional Malaysia (National Defence University of Malaysia), Kem Perdana Sungai Besi, Kuala Lumpur 57000, Malaysia; kibria@upnm.edu.my (G.M.K.); snnl2003@gmail.com (S.N.N.B.L.); nur.syamirah.aishah@gmail.com (N.S.A.B.O.); shahidah2013@gmail.com (S.L.B.A.); 2Department of Basic Health, Kulliyyah of Allied Health Sciences, International Islamic University Malaysia, Jalan Sultan Ahmad Shah, Bandar Indera Mahkota, Kuantan 25200, Malaysia; nazara@iium.edu.my; 3School of Medicine, Swansea University, Swansea, Wales SA2 8PP, UK; j.mckimm@swansea.ac.uk; 4Department of Surgery, Macerata Hospital, University of Macerata, Via Giovanni Mario Crescimbeni, 28, 62100 Macerata MC, Italy; massimosartelli@infectionsinsurgery.org; 5Eastern Medical College, Comilla, Kabila, Dhaka-Chittagong Highway, Burichang 3520, Bangladesh; zakirulislamcom7@gmail.com

**Keywords:** antimicrobial, antibiotic, use, knowledge, attitude, practice, medical students, Malaysia, antimicrobial resistance, antibiotic resistance

## Abstract

**Background**: Antimicrobial prescribing behaviors are often influenced by the local culture and prescribing appropriateness of medical doctors and other health care professionals. Globally, antimicrobial utilization practices have a profound impact on antimicrobial resistance and are a tremendous public health concern. The aim of this survey was to explore the knowledge and attitudes of medical students from the National Defence University of Malaysia regarding antimicrobial usage and antimicrobial resistance. **Research design and methods**: This was a cross-sectional study. The study population consisted of undergraduate medical students in each year group from the National Defence University of Malaysia. Students receive limited formal training on the use of antibiotics in their curriculum, and most of this learning is opportunistic whilst on clinical placement. Universal sampling was used as the study population was small. Data were collected utilizing a previously validated instrument regarding antibiotic use. Simple descriptive statistics were used to generate frequencies and percentages with SPSS V21. This research was approved by the Centre for Research and Innovation Management, National Defence University of Malaysia. **Results**: 206 questionnaires were distributed with a response rate of 99.03%, 54% (110) male, and 46% (94) female. Out of the respondents, 65% (132) had used antibiotics in the last year. Respondents displayed a moderate level of knowledge about antibiotics. **Conclusions**: This study revealed that the older the student was, or when the year of study and total knowledge score was higher, the students were less likely to stop antimicrobials when they felt better or use leftover antibiotics without consulting a doctor. Therefore, the nearer the students were to graduation, the better their knowledge and skills were, and this translated into their own behaviors regarding use of antimicrobials. This finding has clear implications for curriculum design and the inclusion of formal teaching throughout the medical program on antimicrobial use and antimicrobial resistance (AMR). However, more research is needed on this topic, including the prescribing habits and antibiotic use of practicing doctors.

## 1. Introduction

Since the second half of the 19th century, the use of antimicrobial agents has changed the outlook of contemporary medicine, saved an enormous number of people’s lives from deadly microbial infections and relieved misery in patients [1,2,3]. This outstanding success led to a false impression in the late 1960s and early 1970s that infectious diseases had been completely eradicated [4], when, even as late as 2010, an estimated 15 million people died from infections [5]. Furthermore, the World Health Organization (WHO) predicts that there will be 10 million deaths due to antimicrobial resistance (AMR) in 2050 [6]. The rapid emergence of antibiotic resistance is currently a worldwide public health crisis, with a substantial economic and clinical burden on both individual and population health [7,8,9]. The current position regarding antibiotic resistance is often blamed on their overuse and misuse. “Antibiotics have saved countless millions of lives but have been often misused because of the misguided belief that they are harmless” [10].

The way in which doctors prescribe antibiotics and patients adhere to treatment is clearly influential on the rise of AMR, and this is coupled with a lack of new drug development by the pharmaceutical industry, due to reduced economic incentives and challenging regulatory requirements [11,12,13,14,15]. Internationally, both the WHO and the United Nations have identified the global scale of AMR and the need for global strategies [16,17]. The WHO publication “Global Action Plan on Antimicrobial Resistance” [17] sets out “a global consensus that antimicrobial resistance poses a significant public health challenge … emphasizing the paramount significance of achieving the five strategic objectives of the WHO Global Action Plan [17]”. Following the publication of these reports, a high-level meeting on AMR was held, chaired by the President of the United Nations (UN) General Assembly, focusing on how to tackle this public health challenge that has the potential to threaten the health and wellbeing of all people [16].

### 1.1. Prescribing Skills and Medical Students

Prescribing medicines safely and effectively is an essential skill for medical students to learn, because once graduated and licensed to practice as doctors, they will prescribe medicines on a regular basis and habitually with nominal supervision [18,19,20,21]. Good prescribing skills can promote rational and prudent prescribing, which enhances benefits to patients, adherence to treatment and minimizes healthcare costs, however, “antibiotics are misused so often because of the belief that these are benign drugs and that patient satisfaction depends upon being prescribed an antibiotic [18,22]”.

Medical curricula vary in duration, the balance of university-based versus ‘clinical experience’ and graduate outcomes. The way in which pharmacology, clinical pharmacology and prescribing ‘skills’ are taught and learned is also therefore very variable. However, despite these differences, several international studies indicate that medical students lack both the knowledge of infectious diseases and confidence in antibiotic prescribing. One recent study reported that Australian medical students feel less confident in their knowledge of infectious diseases compared to other disease conditions and have inadequate clinical expertise in this area [23]. Another recent study reported that final-year medical students in South Africa possess a low level of confidence regarding antibiotic prescribing, and most students would prefer more education in this area. Overall, knowledge levels were found to be moderate to poor [24]. Multiple European studies conducted among medical students indicate that medical students often lack confidence regarding antimicrobials prescribing as well [18,19,20,21,24,25]. One US study revealed that whilst 92% of medical students thought that adequate knowledge of antimicrobials is essential for doctors, 90% wanted more education regarding their prudent use [26]. Finally, a systematic review of the literature concluded that final-year students had insufficient competencies to prescribe safely and effectively [27].

Because medical programs vary, it is important to contextualize any study within the learning, teaching, and assessment context. At a public defence university in Malaysia, medical students are only exposed to formal lecture-based teaching about antibiotics, antimicrobials, prescribing and AMR in Years 2 and 3 of their studies. These antibiotic related issues were also discussed during clinical case teaching-learning sessions in Years 4 and 5 where relevant. These sessions are opportunistic, and therefore differ between individual students.

### 1.2. Malaysia and Antibiotic Use

Misuse of antimicrobials frequently gives rise to resistant strains of bacteria and is principally related to incorrect prescribing by doctors [28,29,30,31,32,33]. Multiple issues influence the doctors’ choice of antimicrobial (including whether to prescribe them or not) that breach the principles of good clinical practice: for example, the fear of possible future complications in their patients; because of the pressure of patients’ expectations; and a reluctance to potentially damage the doctor-patient relationship [34,35,36,37,38,39].

Multiple studies have been conducted in Malaysia regarding the perception of antibiotic use and resistance (including knowledge, attitude, and practice) among the public, patients, and parents of pediatric patients [40,41,42,43,44,45]. One study was conducted in the six tertiary care hospitals (Malacca, Johor Bahru, Kuantan, Kuala Terengganu, Alor Setar, and Kuching) between August 1991 and July 1992 [46]. This study reported the emergence and rapid increase of antimicrobial resistance against commonly used antibiotics such as ampicillin, cloxacillin, cephalosporins, gentamicin, cotrimoxazole, and tetracycline. Malaysian hospitals are therefore facing a constant threat and challenge to clinicians trying to control nosocomial infections [46]. Educational interventions and training programs have been introduced, aimed at improving the knowledge, attitude, and practice regarding antimicrobials among health care professionals, including medical doctors [1,47,48,49,50,51,52]. Although medical students are the doctors of the future, few studies have been conducted among medical students in Malaysia. This study therefore aimed, for the first time, to explore their usage, knowledge, and attitude regarding antibiotics and AMR and their associated factors such as stopping to take antibiotics once they feel better, keeping leftover antibiotics for future use, and using leftover antibiotics without consulting a doctor.

### 1.3. Pathogenic Microbes, Antimicrobials, Mode of Actions, Antimicrobials Resistance Situations

Microbes are prolific, wide-ranging, adaptable and symbiotic with all mammalian communities. Microbes often cause great harm and are a leading cause of mortality around the world to both human and animal health, which varies according to the host’s immunity level and other environmental factors [53,54]. The invention of the first antimicrobials–including penicillin–was groundbreaking. Over the next two decades, 20 new classes of antimicrobials were developed, including β-lactams, aminoglycosides, tetracyclines, macrolides, fluoroquinolones, cephalosporins, modified β-lactams and β-lactamase inhibitors. These antimicrobials effectively managed even the most dangerous pathogens such as the *Enterobacteriaceae* family [53,55,56,57].

Antimicrobials work by inhibiting vital microbial physiological and biochemical functions, triggering either a bactericidal or bacteriostatic effect by inhibiting the microbial cell wall, the cell membrane, protein, DNA and RNA, and folic acid synthesis [58,59]. Whilst many infectious diseases were being treated effectively, it was felt that microbial diseases would be ultimately eradicated. However, infectious diseases are still the main cause of death globally–especially in lower- and middle-income countries. This is primarily due to infectious diseases, which were once effectively treated but are now returning because of antimicrobial resistance [54]. Microbial resistance is a natural phenomenon, but the overuse and misuse of antimicrobials has led to a serious global crisis [7,60]. The microbial resistant mechanism is naturally developed through chromosomal mutations [59,61] but resistance is acquired as well through extrachromosomal elements from other microbes, including plasmids, bacteriophages, transposons, and integrons [55,62]. Microbes have developed extraordinary mechanisms of resistance to the bacteriostatic or bactericidal effect of antimicrobial molecules, developed over millions of years of successive generations [56,57]. Microbes typically attain resistance by multiple biochemical paths, and one microbial cell can achieve a variety of mechanisms of resistance to ensure their survival against an antimicrobial molecule [59]. So far, scientists have recognized four different types of resistance mechanisms, which are: (i) amendments of the antimicrobial target site, (ii) a diminution in the antimicrobial uptake, (iii) instigation of efflux process to extrude the antimicrobial molecule, or (iv) total deviations in essential metabolic paths via intonation of control systems [59,63]. The consequence is that multidrug-resistant microorganisms are spreading round the world, leading to the rise of both new and previously controllable infectious diseases and a global public health disaster [7,64].

## 2. Methods

### 2.1. Study Design

A cross-sectional design was used in this study to gather information on knowledge regarding antibiotics and their medication among medical students. A previously validated printed survey questionnaire was used to gather data in the most time and cost-efficient way. 

### 2.2. Study Population

The study population comprised Years 1 to 5 medical students from the public defence university in Malaysia. Students are admitted into this university under three categories: cadet officer, territorial army, and civil students.

### 2.3. Study Period

The data were collected from 1 January to 30 April, 2018.

### 2.4. Sampling Method and Sample Size

The total population sampling method was adopted in this study, as the total number of students in the study population was quite small (*n* = 231).

### 2.5. Techniques of Data Collection

Data were collected using a validated 31-point self-completed questionnaire survey, comprising five sections regarding antibiotic use [65]; see Appendix A. Permission was obtained from the primary researcher.Section 1: To gather socio-demographic data, including age, gender, year of birth, religion, year of study and type of admission.Section 2: To evaluate the students’ use of antibiotics over the previous year.Section 3: To assess knowledge regarding antibiotic and related adverse reactions.Section 4: To gather information about respondents’ awareness of antibiotic resistance.Section 5: Focusing on attitudes and behaviors towards antibiotic use.

Responses were scored against a four-point Likert-scale (“Strongly disagree” to “Strongly agree”) and dichotomous answers (yes/no) [66]. Prior to the full survey, the questionnaire was pre-tested and validated in the local context. It was administered to 25 medical students (five students from each year of study) who did not participate in the primary study. Most of the sections of this questionnaire demonstrated acceptable values of Cronbach alpha, with a range between 0.672 and 0.882, which indicated that the instrument possessed good internal consistency and reliability. The evidence of convergent validity was shown by the significant correlations between the items of each section and the total mean in each section (*r*-values = 0.332–0.718; *p* < 0.05) [67,68]. Following pre-testing, 206 questionnaires were distributed among the medical students of the public defence university from Years 1 to 5 in the 2017–2018 academic year.

### 2.6. Statistical Analysis

SPSS V21 (IBM Corporation, Armonk, NY, USA) was used to analyze the data. Frequencies and percentages were used to describe the sociodemographic variables and use of antibiotics among the respondents, and to describe the knowledge, awareness, and attitude of the respondents regarding antibiotics and their resistance. By setting the significance level at 0.05 for 95% confidence interval, a Chi-square test was used to explore the factors associated with the usage of antibiotics by comparing those who used and did not use antibiotics. In addition, a Pearson correlation test was used to explore the association between total knowledge and attitude scores with numerical and ordinal sociodemographic variables—namely age, year of study and grade. An independent *t*-test was used to compare the total knowledge and attitude scores between two independent groups, namely gender, and whether the respondents had any relatives working in health sector. Finally, simple logistic regression was used to assess the factors associated with antibiotics consumption—namely stopping antibiotics when feeling better, keeping leftover antibiotics for future use and using leftover antibiotics without doctor’s consultation.

### 2.7. Ethical Considerations

This research study was approved by the Institutional Ethical Clearance Committee, Centre for Research and Innovation Management, National Defence University of Malaysia (UPNM/2017/SF/SKK/04, memo number: UPNM (PPPI) 16.01/06/024 (2), dated 23 August, 2017). The study population was informed about the objectives and process of the study, that the data gathered would be anonymized and used for publication and that the study participation was totally voluntary. Written consent was then obtained before the questionnaires were distributed.

## 3. Results

### 3.1. Socio-Demographic Profile of the Study Respondents

From the 206 questionnaires distributed, 204 were returned, which meant a response rate of 99.0%. There were slightly more male respondents (*n* = 110, 53.9%) compared to females. The majority of the respondents were Muslims (*n* = 140, 68.9%) and did not have any relatives working in the health field (*n* = 147, 72.1%). Most respondents (56 respondents, 27.5%) were from Year 1 of the program with the other years of study almost similarly distributed, as shown in Table 1.

The mean age of the respondents was 21.8 years old (standard deviation, SD = 1.49), with age highly corresponding with the year of study (Year 1 = 20 years old; Year 2 = 21 years old; Year 3 = 22 years old; Year 4 = 23 years old and Year 5 = 24 years old), except for three Year 4 students aged 24 years old and two Year 5 students aged 23 and 25 years’ old each. Almost half of the respondents were admitted as a cadet officer (*n* = 100, 49.3%), and the most common latest academic grade obtained by the respondents were B (*n* = 150, 73.5%).

### 3.2. Usage of Antibiotics

From the total 204 respondents, more than half (*n* = 133, 65.2%) had used antibiotics in the previous year. From those who had used antibiotics in the previous year, the majority (*n* = 103, 77.4%) had used antibiotics only once or twice, with another 27 (20.3%) and three (2.3%) respondents having used them three to five or more than five times, respectively.

### 3.3. Knowledge, Awareness, and Attitude Regarding Antibiotics

Table 2 shows the descriptive analysis for the knowledge regarding antibiotics among the respondents. Generally, their perceived knowledge was good with 180 (more than 88%) respondents totally agreeing or agreeing with the statements: “Penicillin or Amoxicillin are antibiotics,” “Antibiotics are useful for bacterial infections” and “Antibiotics can cause allergic reactions.” Similarly, the majority totally disagreed or disagreed with the statements of “Aspirin is an antibiotic” (*n* = 166, 81.4%), “Paracetamol is an antibiotic” (*n* = 173, 84.8%), “Antibiotics are useful for viral infections” (*n* = 134, 65.6%) and “Antibiotics are indicated to reduce any kind of pain and inflammation” (*n* = 144, 70.6%).

Furthermore, responses showed a reasonably high degree of awareness about antibiotic resistance among the respondents, with 87.7% (*n* = 179) stating that they had heard about it, either through their degree course (*n* = 138, 67.6%) or from outside the course (*n* = 153, 75.0%). Figure 1 shows the sources outside the degree course from which the responses got the information regarding antibiotic resistance.

In addition, more than three quarters of the respondents answered “Totally agree” or “Agree” for the statements: “Antibiotic resistance is a phenomenon for which a bacterium loses its sensitivity to an antibiotic” (*n* = 178, 87.3%) and “Misuse of antibiotics can lead to a loss of sensitivity of an antibiotic to a specific pathogen” (*n* = 175, 85.8%), while 159 (78.0%) answered “Totally disagree” or “Disagree” for the statement: “If symptoms improve before it is completed the full course of antibiotic, you can stop taking it” (Table 3).

Table 4 sets out the responses regarding attitudes regarding personal use of antibiotics. The respondents’ correct answers for various statements were: “Do you usually take antibiotics for a cold or a sore throat?” (77.5%); “Do you usually take antibiotics for fever?” (60.8%); “Do you usually stop taking antibiotics when you start feeling better?” (67.2%); “Do you take antibiotics only when prescribed by the doctor?” (87.7%); “Do you keep leftover antibiotics at home because they might be useful in the future?” (66.7%); “Do you use leftover antibiotics when you have cold, sore throat or flu without consulting your doctor?” (77.5%); “Do you buy antibiotics without a medical receipt?” (88.7%), and “Have you ever started an antibiotic therapy after a simple doctor call, without a proper medical examination?” (83.3%) 

### 3.4. Factors Associated with Usage of Antibiotics

A Chi-square test was used to identify the factors associated with the use of antibiotics in the previous years among the respondents. Table 5 shows that for the comparison of sociodemographic variables between those who used and did not use antibiotics, there were significant results—except for the year of study (*p* = 0.024) and type of admission (*p* = 0.050—which were borderline significant.

Bar charts were used to further clarify the significant differences seen in terms of year of study and types of admission, as shown in Figure 2 and Figure 3, respectively. Figure 1 clearly shows these differences in terms of year of study, the use of antibiotic in the previous year (the blue bar noted as “yes”) was higher in the lower years of study—especially in Years 1, 2 and 3. Figure 2 shows the differences in terms of the type of admission, where it can be seen that the use of antibiotic in the previous year (the blue bar noted as “yes”) was higher among the respondents admitted as cadet officers and civil students as compared to as territorial army students.

### 3.5. Factors Associated with Total Knowledge and Attitude Scores Regarding Antibiotics

To assess the factors associated with total knowledge and attitude scores regarding antibiotics, the total knowledge scores were calculated by adding the scoring from all questions from Section 3 (on knowledge) and the last three questions from Section 4 (on awareness), while the total attitude scores were the total of scoring from all questions from Section 5 (on attitude). The negative questions were given a reverse scoring.

A Pearson correlation test shows statistically significant moderate to a good positive correlation between total knowledge scores with age and year of study (*p* < 0.001), as well as a statistically significant fair, positive correlation between total attitude scores with age and year of study (*p* < 0.001), as shown in Table 6. However, no significant correlation was found between total knowledge and attitude scores with examination grade.

Using the same total knowledge and attitude scores as mentioned above, an independent *t*-test shows no statistically significant difference regarding total knowledge and attitude scores between male and female respondents or whether they had relatives working in the health field or not (Table 7).

### 3.6. Factors Associated with Stopping Antibiotics When Feeling Better, Keeping Leftover Antibiotics for Future Use and Using Leftover Antibiotics without Doctor’s Consultation

A simple logistic regression found that when students were older, or when their year of study and total knowledge scores were higher, the odds were lower for the respondents to stop taking antibiotics when they feel better (OR = 0.782, *p* = 0.019; OR = 0.767, *p* = 0.013; OR = 0.906, *p* < 0.001; respectively) compared to younger students or those with lower year of study and total knowledge scores, respectively. Similar results were seen regarding using leftover antibiotics without a doctor’s consultation (OR = 0.767, *p* = 0.028; OR = 0.769, *p* = 0.030; OR = 0.880, *p* < 0.001; respectively). In addition, those who had heard about antibiotic resistance were at higher odds of stopping antibiotics when they feel better (OR = 2.508, *p* = 0.033) compared to those who had not heard of antibiotic resistance. However, only the total knowledge score was still significant at a multivariate level for the first two outcomes mentioned above (OR = 0.924, *p* = 0.018 and OR = 0.879, *p* = 0.001, respectively). None of the variables shows a significant association with keeping the leftover antibiotics for future use (*p* = 0.209 to 1.000). The detailed results are summarized in Table 8.

## 4. Discussion

Antimicrobials have played an essential role in progressing modern medical and surgical care, minimizing the global burden of communicable disease, and extending human life expectancy [69]. These medicines have transformed modern medical practice, but the persistent evolutionary changes of microorganisms resulting in antimicrobial resistance have become a major global public health threat [70]. “In the absence of urgent corrective and protective actions, the world is heading towards a post-antibiotic era, in which many common infections will no longer have a cure and, once again, kill unabated” [71]. The primary objective of the current study was to study the knowledge and attitudes of medical students in a public defence university in Malaysia towards antibiotics, as research on this in the Malaysian context is limited. To the best of the current researchers’ knowledge, only one paper had been published regarding the prescribing, use of, and resistance to antimicrobials among medical students in Malaysia [20].

### 4.1. Socio-Demographic Profile of the Study Respondents

The current study response rate was 99.03%. A similar high response rate was also reported in earlier studies [72,73]. The American Journal of Pharmaceutical Education reported that in “survey research intended to represent all schools and colleges of pharmacy, a response rate of ≥80% is expected” [74]. The current study response rate can therefore be considered good. In the current study, there was a slightly higher number of male respondents, again, similar to earlier studies [72,73] but dissimilar to other Malaysian medical survey studies [75,76]. The context being a Malaysian military medical school most likely influenced the gender balance due to a high number of cadet officers. How these characteristics influenced their knowledge and views are still to be determined. Malaysia is a predominantly Malay and Muslim country, and the current study respondents reflected this—being principally Malay and Muslim. These cultural factors most likely influenced the responses (e.g., knowledge of antibiotic resistance obtained from general media) but were not directly explored in this study.

### 4.2. Use of Antibiotics and Its Associated Factors

Antibiotics are one of the most commonly prescribed groups of medicines but are widely misused with the consequent development of AMR. AMR increases the health care burden by increasing morbidity and mortality due to infections developing resistant strains. The problem of AMR has a global dimension, despite the WHO’s best efforts to raise awareness of this challenge [77]. The issue is therefore equally important for emerging and technologically advanced countries to address.

In the current study, most respondents had used antimicrobials on one to two occasions in the previous year, which is higher than in an earlier European study [65]. One recent Chinese study of among medical students found 54% of respondents had self-prescribed antibiotics, 64% had stocked antibiotics, and 58% had been prescribed inappropriate antibiotics by doctors [78]. This was not confined to medical students, for example a Sri Lankan study of health sciences students reported that overall, 39% of students had self-medicated antibiotics with (interestingly) the percentage of antibiotic self-medication being statistically significantly (*p* = 0.001) higher in the pharmacology education group than in the non-formal pharmacology education group [79]. A statistically significant (*p* < 0.001) correlation was observed as well with a year of study. Another, very recent Sri Lankan study revealed that the majority (75%) of pharmacy students took antimicrobials during the previous year and procured antimicrobials (77%) by prescription [80].

The current study found that Year 1 students had higher antibiotic use than the rest of the years. However, the Sri Lankan studies had exactly the opposite findings [79,80]. The findings in the current study could be because Year 1 students were leaving their parents for the first time and learning to make their way in life without direct support and guidance, where they lived in crowded hostels and were exposed to new infections. As students’ progress through their medical program, the propensity of common illnesses could be reduced, and they are learning more about how to use antibiotics.

### 4.3. Knowledge, Awareness, and Attitude Regarding Antibiotics and Its Associated Factors

“Knowledge is a structural property of attitudes that is a function of the number of beliefs and experiences linked to the attitude in memory and the strength of the associative links between the beliefs or experiences and the attitude” [81]. Multiple studies report that an increase in knowledge is often associated with greater behavioral changes—especially in health-related issues [82,83,84,85,86].

#### 4.3.1. Knowledge Regarding Antibiotics

The current study respondents possessed an average level of knowledge regarding antibiotics, with 68% of them correctly answering the knowledge related questions. This finding is lower than that of an earlier Italian study, where the recorded correct responses were 90% [7]. However, 34.3% respondents of the current study believed that antibiotics were useful in viral diseases and 56.4% believed that they could cause secondary infections, respectively. These two figures are higher than in the earlier Italian study mentioned above [7]. AMR is a global crisis threating modern medical science because of the constant high-speed spread of antimicrobial-resistant microorganisms [87]. Moreover, the development of new antibiotics is slow [88] because many leading research-based pharmaceutical industries have stopped research programs aimed at developing new antimicrobials [89,90,91]. For example, the US Food and Drug Administration (FDA) approved only one-tenth of new antimicrobials in the last 35 years compared with the previous 35 years [92,93]. Briefly, “no approved antibiotic drug class has been discovered since 1980. Of even greater concern is the fact that no new class of antibiotics has been discovered to treat Gram-negative bacteria since 1962” [94].

It seems apparent that almost no new antimicrobials will appear on the market over the next few years. Educational interventions both in undergraduate studies and continuous medical education regarding antibiotic use and prudent prescribing are therefore essential to influence prescribing behavior by providing updated information and knowledge and improve the acceptance of stewardship approaches [95]. As AMR is a global crisis, a recent review suggested that coordinated policy and planning is needed to make the best use of resources to develop new antimicrobials, as well as to allow their best utilization [96].

#### 4.3.2. Awareness of Antibiotic Resistance

AMR is an international issue underpinning and influencing the treatment of both primary and secondary infectious diseases. It is essential therefore that doctors are educated properly so that they acquire the requisite knowledge, are trained to prescribe appropriately (including not prescribing antibiotics if it is not indicated) and have the professional and communication skills to champion the best practice. Awareness about AMR hinges not just on doctors but on other health professionals, for example, “the Global Action Plan on AMR highlights the importance of training all healthcare professionals” [97], and the attitudes towards and knowledge about AMR of patients [98].

The current study showed that the respondents answered about 80% of AMR awareness questions correctly—displaying good awareness about AMR—which is similar to respondents in a recent Romanian study which found that 84% of their medical residents demonstrated awareness of AMR [98]. It is important to notice, however, that the Romanian study population consisted of graduate medical doctors, and the current study respondents were medical students. Nevertheless, in the current study, only 60% totally agreed that the misuse of antibiotics could lead to AMR, whereas one research survey among medical students in the United States reported that almost all their students were aware that inappropriate use of antimicrobials could cause AMR [26]. This variation in awareness clearly reflects the way in which this topic is taught in different curricula.

#### 4.3.3. Attitudes Regarding Personal Consumption of Antibiotics

Medical students are going to be practicing medical doctors soon after graduation. It was observed in Hong Kong that medical students who had received clinical training were more aware of AMR, and after graduation prescribed antibiotics more rationally than those who had not received clinical training, and thus did not acquire the correct knowledge of antibiotic use [99]. The study concluded that developing sound knowledge and appropriate behaviors regarding AMR and antimicrobial use are important factors to regulate physician stewardship in antimicrobial prescribing [99]. The current study respondents correctly answered the attitude related questions on an average of 66%, compared with 79% in the earlier Italian study [65], which reflects that more consistent training is most likely required.

Most of the current study respondents did not take antimicrobials for the common cold, sore throat, and fever—unlike one recent survey in Poland among the public, which found that the main reasons for taking antimicrobials were the common cold, sore throat, cough, and flu. Nearly 40% of the study subjects in Poland expected a prescription for antimicrobials against flu [100]. Another study reported that the general population in Serbia thought that antibiotics were useful in treating the common cold [101]. Almost half of the respondents in that study took antibiotics without a doctor’s prescription at least once during their lifetime and during the previous treatment of infection [101].

On the other hand, among the current study respondents, 32.8% stop taking antibiotics when they start feeling better, and 33.3% preserved leftover antimicrobials for future use. Responses to a similar question in an Indian study found responses of 21.6% and 37%, respectively, and additionally, 21.6% shared leftover medicine with their friends or roommates when they were sick [102]. A similar figure was observed regarding consulting a doctor before starting antimicrobials in both the Indian (90.8%) [102] and in this current (87.7%) study.

#### 4.3.4. Factors Associated with Knowledge, Awareness, and Attitude Regarding Antibiotics

The study found no significant (*p* > 0.05) correlation with gender, race, having a close relative working in a health-related field, and latest academic grade regarding antibiotic use within last one year. Nevertheless, a significant correlation was observed in the year of study (*p* = 0.024) and type of admission (*p* = 0.050). Moreover, antibiotic consumption was observed more frequently among junior medical students and cadet officers. This differed from an earlier study, which reported that females and those with relatives working in a health-related field had consumed significantly more antibiotics than the comparison group [65]. As medical students progress in the program and start clerking patients, they seem to better understand the role of antibiotics and abuse in practice and, their own consumption of antibiotics reduces.

### 4.4. Factors Associated with Stopping Antibiotics When Feeling Better, Keeping Leftover Antibiotics for Future Use and Using Leftover Antibiotics without Doctor’s Consultation

The WHO advises patients to “always complete the full prescription, even if you feel better because stopping treatment early promotes the growth of drug-resistant bacteria” [103]. Similar recommendations have been made regarding completing a whole course of antimicrobials by multiple national and international guidelines and protocols [104,105,106]. “However, the idea that stopping antibiotic treatment early encourages antibiotic resistance is not supported by evidence while taking antibiotics for longer than necessary increases the risk of resistance” [104]. The current study revealed that when students are older—or when the year of study and total knowledge score is higher—the lower the odds are of the students stopping antibiotics when they feel better or using leftover antibiotics without doctor’s consultation. This may be a coincidence with very recent developments reported by Llewelyn and colleagues in the British Medical Journal but may be considered as demonstrating evidence of a good positive attitude towards AMR [107]. Moreover, the WHO is currently appraising its guidelines on prudent antimicrobial prescribing [103].

## 5. Limitations

This is a cross-sectional study with its inherent limitations. The study population, and thus, the data set, is small as there are only 50 students in each year cohort: a total study population of 231. Additionally, the instrument adopted was a self-administered questionnaire based on self-reporting, which can lead to recall bias [108]. This study was conducted in only one center based on students studying one medical curriculum; therefore, the current findings are difficult to generalize.

## 6. Conclusions

Being able to prescribe medicines safely and effectively is an essential skill of medical graduates. Most respondents in this study possess a good level of knowledge and took antimicrobials only when prescribed. The nearer the students were to graduation, the better their knowledge and skills were, and this translated into their own behaviors regarding use of antimicrobials. The Year 1 students used antimicrobials more frequently than students from the later years. No statistically significant differences were observed between sexes. The current study findings provide baseline data for future in-depth research studies to design a more appropriate curriculum on antibiotic use for medical students and practicing doctors in Malaysia and elsewhere. These findings have clear implications for curriculum design and the inclusion of formal teaching throughout the medical program on antimicrobial use and AMR. However, more research is needed on this topic, including the prescribing habits and antibiotic use of practicing doctors. Further research should explore the impact of curriculum interventions on knowledge, attitudes and clinical practice, vitally important as guidance on antibiotic use is changing, and doctors need to be aware of these shifts in practice as they are at the forefront of prescribing drugs to their patients.

## Figures and Tables

**Figure 1 antibiotics-08-00154-f001:**
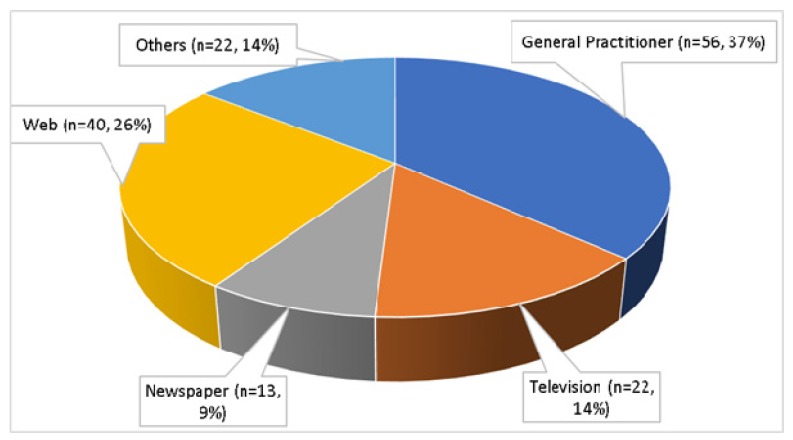
The sources of information on antibiotic resistance besides formal teaching (*n* = 153).

**Figure 2 antibiotics-08-00154-f002:**
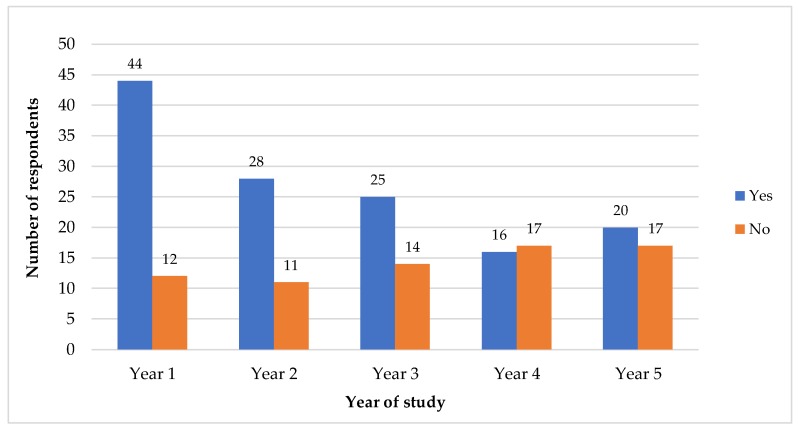
Distribution of use of antibiotic in the previous year within different years of study (*n* = 204).

**Figure 3 antibiotics-08-00154-f003:**
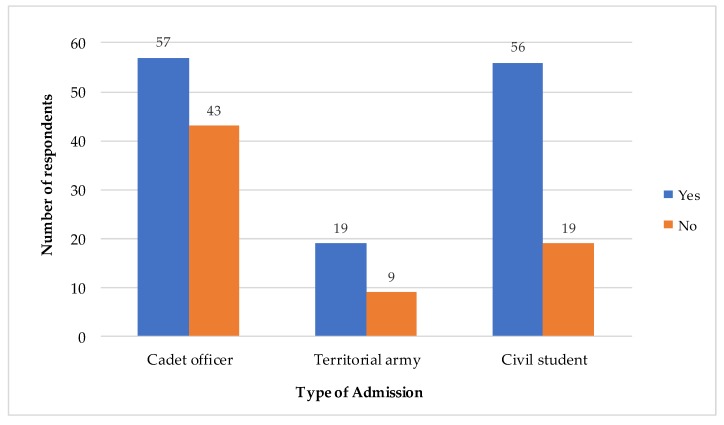
Distribution of use of antibiotic in the previous year within different types of admission (*n* = 204).

**Table 1 antibiotics-08-00154-t001:** Distribution of the respondents’ socio-demographic factors (*n* = 204).

	Frequency	Percentage
**Gender**		
Male	110	53.9
Female	94	46.1
**Race**		
Malay	140	68.9
Chinese	18	8.8
Indian	41	20.1
Others	5	2.5
**Had relatives working in health field**		
Yes	57	27.9
No	147	72.1
**Year of study**		
1	56	27.5
2	39	19.1
3	39	19.1
4	33	16.2
5	37	18.1
**Type of admission ^a^**		
Cadet officer	100	49.3
Territorial army	28	13.8
Civil student	75	36.9
**Latest academic grade**		
A	20	9.8
B	150	73.5
C	34	16.7

^a^ with one missing value.

**Table 2 antibiotics-08-00154-t002:** Knowledge about antibiotics (*n* = 204).

Questions on Knowledge about Antibiotic	Totally Disagree *n* (%)	Disagree *n* (%)	Agree *n* (%)	Totally Agree *n* (%)
● Penicillin or Amoxicillin are antibiotics ^a^	9 (4.4)	14 (6.9)	31 (15.3)	149 (73.4)
● Aspirin is an antibiotic	142 (69.6)	24 (11.8)	29 (14.2)	9 (4.4)
● Paracetamol is an antibiotic	157 (77.0)	16 (7.8)	15 (7.4)	16 (7.8)
● Antibiotics are useful for bacterial infections (e.g., tuberculosis)	11 (5.4)	13 (6.4)	31 (15.2)	149 (73.0)
● Antibiotics are useful for viral infections (e.g., flu)	108 (52.9)	26 (12.7)	29 (14.2)	41 (20.1)
● Antibiotics are indicated to reduce any kind of pain and inflammation	102 (50.0)	42 (20.6)	37 (18.1)	23 (11.3)
● Antibiotics can kill “good bacteria” present in our system	29 (14.2)	53 (26.0)	54 (26.5)	68 (33.3)
● Antibiotics can cause secondary infections after killing good bacteria present in our system	26 (12.7)	63 (30.9)	59 (28.9)	56 (27.5)
● Antibiotics can cause allergic reactions	5 (2.5)	19 (9.3)	57 (27.9)	123 (60.3)

^a^ with one missing value.

**Table 3 antibiotics-08-00154-t003:** Awareness of antibiotic resistance (*n* = 204).

	Totally Disagree *n* (%)	Disagree *n* (%)	Agree *n* (%)	Totally Agree *n* (%)
● Antibiotic resistance is a phenomenon for which a bacterium loses its sensitivity to an antibiotic.	6 (2.9)	20 (9.8)	41 (20.1)	137 (67.2)
● Misuse of antibiotics can lead to a loss of sensitivity of an antibiotic to a specific pathogen.	5 (2.5)	24 (11.8)	52 (25.5)	123 (60.3)
● If symptoms improve the full course of antibiotics is completed, you can stop taking it.	137 (67.2)	22 (10.8)	25 (12.3)	20 (9.8)

**Table 4 antibiotics-08-00154-t004:** Attitudes regarding personal consumption of antibiotics (*n* = 204).

Questionnaire on Attitudes Regarding Antibiotic Consumption	Yes *n* (%)	No *n* (%)
● Do you usually take antibiotics for cold or sore throat?	46 (22.5)	158 (77.5)
● Do you usually take antibiotics for fever?	80 (39.2)	124 (60.8)
● Do you usually stop taking antibiotics when you start feeling better?	67 (32.8)	137 (67.2)
● Do you take antibiotic only when prescribed by the doctor?	179 (87.7)	25 (12.3)
● Do you keep leftovers antibiotics at home because they might be useful in the future?	68 (33.3)	136 (66.7)
● Do you use leftovers antibiotics when you have a cold, sore throat or flu without consulting your doctor?	46 (22.5)	158 (77.5)
● Do you buy antibiotics without a medical receipt?	23 (11.3)	181 (88.7)
● Have you ever started an antibiotic therapy after a simple doctor call, without a proper medical examination?	34 (16.7)	170 (83.3)

**Table 5 antibiotics-08-00154-t005:** Chi-square test on factors associated with the usage of antibiotics (*n* = 204).

Variables	Use Antibiotic in the Previous Year		Chi-Square Value (df ^a^)	*p*-Value
	Yes *n* (%)	No *n* (%)		
Gender				
Male	75 (56.4)	35 (49.3)	0.938	0.333
Female	58 (43.6)	36 (50.7	(1)	
Race				
Malay	88 (66.2)	52 (73.2)	1.076	0.300
Others ^b^	45 (33.8)	19 (26.8)	(1)	
Had relatives working in the health field				
Yes	39 (29.3)	18 (25.4)	0.363	0.547
No	94 (70.7)	53 (74.6)	(1)	
Year of study				
1	44 (33.1)	12 (16.9)	11.270	0.024
2	28 (21.1)	11 (15.5)	(4)	
3	25 (18.8)	14 (19.7)		
4	16 (12.0)	17 (23.9)		
5	20 (15.0)	17 (23.9)		
Type of admission ^c^				
Cadet officer	57 (43.2)	43 (60.6)	5.996	0.050
Territorial army	19 (14.4)	9 (12.7)	(2)	
Civil student	56 (42.4)	19 (26.8)		
Latest academic grade				
A	13 (9.8)	7 (9.9)	2.770	0.250
B	102 (76.7)	48 (67.6)	(2)	
C	18 (13.5)	16 (22.5)		

^a^ df = degree of freedom; ^b^ Chinese, Indian and other races; ^c^ with one missing value.

**Table 6 antibiotics-08-00154-t006:** Association between total knowledge and attitude scores with age, year of study, and examination grade using the Pearson correlation test (*n* = 204).

Variables	Total Knowledge Score		Total Attitude Score	
	*r*-Value	*p*-Value	*r*-Value	*p*-Value
Age	0.568	<0.001	0.252	<0.001
Year of Study	0.572	<0.001	0.258	<0.001
Grade	0.038	0.591	−0.112	0.109

**Table 7 antibiotics-08-00154-t007:** Comparing total knowledge and attitude scores between different genders and having relatives working in the health field using independent *t*-test (*n* = 204). * SD = standard deviation.

Variables	Total Knowledge Score		Total Attitude Score	
	Mean (SD *)	*p*-Value	Mean (SD)	*p*-Value
Gender				
Male (*n* = 110)	39.7 (5.59)	0.598	14.0 (1.89)	0.348
Female (*n* = 94)	39.2 (6.43)		14.2 (1.75)	
Had relatives working in health field				
Yes (*n* = 57)	39.7 (5.50)	0.688	14.1 (1.81)	0.819
No (*n* = 147)	39.4 (6.17)		14.1 (1.84)	

**Table 8 antibiotics-08-00154-t008:** Simple logistic regression on factors associated with antibiotic consumption (*n* = 204).

Variables	Stopping Antibiotic when Feeling Better		Keeping Leftover Antibiotics for Future Use		Using Leftover Antibiotics without a Doctor’s Consultation	
	OR (95% C.I.)	*p*-Value	OR (95% C.I.)	*p*-Value	OR (95% C.I.)	*p*-Value
Age *	0.782 (0.636, 0.961)	**0.019**	0.929 (0.762, 1.132)	0.463	0.767 (0.606, 0.971)	**0.028**
Year of study *	0.767 (0.622, 0.946)	**0.013**	0.926 (0.758, 1.132)	0.456	0.769 (0.606, 0.976)	**0.030**
Exam grade *	1.228 (0.692, 2.181)	0.483	1.447 (0.813, 2.576)	0.209	1.515 (0.793, 2.894)	0.208
Total knowledge Score *	0.906 (0.860, 0.954)	**<0.001 ****	0.972 (0.926, 1.021)	0.257	0.880 (0.829, 0.934)	**<0.001 ****
Gender						
Male ^#^	1	1.000	1	1.000	1	1.000
Female	0.827 (0.460, 1.485)	0.525	0.789 (0.440, 1.415)	0.427	1.622 (0.826, 3.185)	0.160
Relatives working in health-related field						
No ^#^	1	1.000	1	1.000	1	1.000
Yes	1.533 (0.777, 3.024)	0.218	0.806 (0.425, 1.528)	0.508	0.855 (0.416, 1.754)	0.669
Heard about antibiotic resistance						
No ^#^	1	1.000	1	1.000	1	1.000
Yes	2.508 (1.075, 5.850)	**0.033**	1.391 (0.589, 3.284)	0.452	2.159 (0.884, 5.274)	0.091
Discussed about antibiotic resistance						
No ^#^	1	1.000	1	1.000	1	1.000
Yes	1.538 (0.832, 2.842)	0.170	1.000 (0.537, 1.863)	1.000	1.662 (0.843, 3.277)	0.142

* Treated as continuous variables; OR = Odds Ratio; C.I. = Confidence Interval; ** Significant at a multivariate level using multiple logistic regression; ^#^ Control group. Bold is significant.

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
