# Peer review of "Antibiotic Use: A Cross-Sectional Study Evaluating the Understanding, Usage and Perspectives of Medical Students and Pathfinders of a Public Defence University in Malaysia"

_antibiotics, 2019, doi:10.3390/antibiotics8030154_

Round 1

Reviewer 1 Report

General comments to the Authors

In this manuscript the authors sought to explore antimicrobial prescribing behaviors are often influenced by the local culture and prescribing appropriateness of medical doctors and other health care professionals among medical students in Malaysia. This study therefore aimed (for the first time) to estimate their usage, knowledge, and attitude regarding antibiotics and AMR and their associated factors, the factors associated with stopping taking antibiotics once they feel better, keeping leftover antibiotics for future use, and using leftover antibiotics without consulting a doctor. In this model, the data were collected utilizing a validated instrument regarding antibiotic use. Finally, this study revealed that the older the student was, or when the year of study and total knowledge score was higher, the students were less likely to stop antimicrobials when they felt better or use leftover antibiotics without consulting a doctor. Overall it is not a good written manuscript that the presences of the inclusion of formal teaching throughout the medical program on antimicrobial use and antimicrobial resistance. Furthermore, there are some points the need further clarification as follows.

Abstract: this part is digression, especially in logically arranged descriptions. It is need to be organized. Introduction: the description are occupied too much spaces; these authors should be written more concisely. The paragraphs are described a little digression and these authors should be written more concisely. Methods: The analytical methods is too rough. I suggest that the authors need to show the more and clear references of hypothesis testing for this model significant. Results: this part is need to be reorganization and written more concisely. Many descriptions in this part are like the parts of Discussion. In the Figure 3, what the authors want to show, especially in [] (n=[], [])? Discussion: This part was poorly described. This part need more information to show the significant among different countries or references. Conclusions: these descriptions of this part is not different from Introduction or Discussion. I am not sure the authors wanted to show the “Introduction”, “discussion” or “Conclusions”.

Author Response

Reviewer 1

Abstract: This part is digression, especially in logically arranged descriptions. It needs to be organized. 

The abstract has been edited accordingly.

Introduction: The description is occupied too much spaces; these authors should be written more concisely. The paragraphs are described a little digression and these authors should be written more concisely. 

We have tried to alter as possible.  Another reviewer-II asks to add a new section on top of present Introduction

Methods: The analytical methods are too rough. I suggest that the authors need to show the more and clear references of hypothesis testing for this model significant. 

The statistical methods used were described in more detail under the Data Analysis section, which has been renamed as “Statistical Analysis”.

Results: this part needs to be reorganization and written more concisely. Many descriptions in this part are like the parts of Discussion.

This part had been re-arranged, by presenting the descriptive results first (3.1 – 3.3), followed by the results on associated factors with further comparisons (3.4 – 3.6). The explanation has also been added to make the interpretation clearer, especially on the comparison parts (3.4 – 3.6).

In the Figure 3, what the authors want to show, especially in [] (n=[], [])? 

Figure 3 has been corrected. The wordings and numbers are now visible. It shows the sources of information on antibiotics besides the formal teaching (the title has also been corrected).

Discussion: This part was poorly described. This part needs more information to show the significant among different countries or references. 

We have tried to alter as possible.

Conclusions: These descriptions of this part is not different from Introduction or Discussion. I am not sure the authors wanted to show the “Introduction”, “discussion” or “Conclusions”. 

The content has been edited to clearly showed the conclusion of the study, and the title was amended to “Conclusion and Recommendations” to better represent the content of the paragraph.

Reviewer 2 Report

The manuscript describes the antibiotic usage and understanding its resistance from the medical students in the various groups (from first to four years) from the National Defence University of Malaysia. The manuscript was well written and organized. This study shows that medical students have relatively good knowledge about antibiotics and take when prescribed by doctors. However, the authors also emphasized a few older students were scored higher and less likely to stop antimicrobial when they felt better or use leftover antibiotics without consulting a doctor. This manuscript is an exciting piece of work and provides a better understanding of the medical student about medicines during graduation. Still, they need more information about the antimicrobial and their mode actions for various microorganisms.  

Author Response

Still, they need more information about the antimicrobial and their mode actions for various microorganisms.  

A new section added. 1.c.

Reviewer 3 Report

Dear Authors,

After the review process, I have several comments:

1. you should eliminate any materials & methods from results section of the abstract;

2. you should include a statistical section in Methods;

3. you should include a clear statistical data & explanation in Results section;

4. you should present which is the control group, because it is an important aspect for the results interpretation;

5. you should comment their results based on the alternatives to antimicrobial products and whether the test group ingested this type of products.

Best regards!

Author Response

Reviewer 3

you should eliminate any materials & methods from results section of the abstract;

The abstract has been edited accordingly.

you should include a statistical section in Methods;

Section 2.f has been renamed as “Statistical Analysis” and the analysis done was described in detail.

you should include a clear statistical data & explanation in Results section;

This part had been re-arranged, by presenting the descriptive results first (3.1 – 3.3), followed by the results on associated factors with further comparisons (3.4 – 3.6). The explanation has also been added to make the interpretation clearer, especially on the comparison parts (3.4 – 3.6).

you should present which is the control group, because it is an important aspect for the results interpretation;

The control group was presented in Table 2 (do not use antibiotics in comparison to those who used antibiotics). In Table 8, the control group is now noted as ‘#’, ie the group with OR=1, and explanation has been improved by including the mention of the control group.

you should comment their results based on the alternatives to antimicrobial products and whether the test group ingested this type of products.

This study only focused on the usage of antibiotics, and not the alternatives. But the point is taken, and we may include this suggestion in our future study.

Round 2

Reviewer 1 Report

In the Figure 3, the wordings and numbers are not now visible in the revision. 

Reviewer 3 Report

Dear Authors,

I do not have supplementary comments. 

Best regards!